# The Timely Needs for Infantile Onset Pompe Disease Newborn Screening—Practice in Taiwan

**DOI:** 10.3390/ijns6020030

**Published:** 2020-04-01

**Authors:** Shu-Chuan Chiang, Yin-Hsiu Chien, Kai-Ling Chang, Ni-Chung Lee, Wuh-Liang Hwu

**Affiliations:** 1Department of Medical Genetics, National Taiwan University Hospital, Taipei 100, Taiwan; chiangsc@ntu.edu.tw (S.-C.C.); bunnyhead198134@gmail.com (K.-L.C.); ncleentu@ntu.edu.tw (N.-C.L.); hwuwlntu@ntu.edu.tw (W.-L.H.); 2Department of Pediatrics, National Taiwan University Hospital, Taipei 100, Taiwan

**Keywords:** infantile-onset Pompe disease, GAA sequencing, immune modulation therapy, enzyme replacement therapy, cross-reactive immunologic material

## Abstract

Pompe disease Newborn screening (NBS) aims at diagnosing patients with infantile-onset Pompe disease (IOPD) early enough so a timely treatment can be instituted. Since 2015, the National Taiwan University NBS Center has changed the method for Pompe disease NBS from fluorometric assay to tandem mass assay. From 2016 to 2019, 14 newborns were reported as high-risk for Pompe disease at a median age of 9 days (range 6–13), and 18 were with a borderline risk at a median age of 13 days (9–28). None of the borderline risks were IOPD patients. Among the 14 at a high-risk of Pompe disease, four were found to have cardiomyopathy, and six were classified as potential late-onset Pompe disease. The four classic IOPD newborns, three of the four having at least one allele of the cross-reactive immunologic material (CRIM)-positive variant, started enzyme replacement therapy (ERT) at a median age of 9 days (8–14). Western Blot analysis and whole gene sequencing confirmed the CRIM-positive status in all cases. Here, we focus on the patient without the known CRIM-positive variant. Doing ERT before knowing the CRIM status created a dilemma in the decision and was discussed in detail. Our Pompe disease screening and diagnostic program successfully detected and treated patients with IOPD in time. However, the timely exclusion of a CRIM-negative status, which is rare in the Chinese population, is still a challenging task.

## 1. Introduction

Pompe disease, a genetic disorder caused by variants of the glucosidase alpha acid (*GAA*) gene, leads from acid alpha-glucosidase (GAA) deficiency. The phenotypes of Pompe disease vary widely, ranging from the most severe classic infantile-onset Pompe disease (IOPD) to the later-onset Pompe disease (LOPD). Currently, enzyme replacement therapy (ERT) with recombinant human GAA (rhGAA) is the only approved therapy. We have performed newborn screening (NBS) for Pompe disease [1] since 2005, and our results demonstrate that the early initiation of treatment improves the prognosis IOPD patients [2], thus confirming the value of newborn screening for Pompe disease. 

However, there are still challenges when an infant receives a positive screening result. First, the phenotype cannot be predicted by GAA activity [3,4]. Mutation analysis of the *GAA* gene can predict the phenotype in a portion of patients [5]. But more precisely, newborns with classic IOPD should have presented with cardiomyopathy and muscle weakness at birth clinically [2,6]. Second, ERT with rhGAA may trigger an immune response with neutralizing antibodies, especially in patients negative for the cross-reactive immunologic material (CRIM) [7]. Prophylactic immunologic modulation therapy may overcome the problem, but the CRIM status needs to be defined before initiating ERT [8]. Nowadays, some GAA variants are associated with a known CRIM status [9]. 

The National Taiwan University Hospital (NTUH) Newborn Screening Center, established since 1985, is responsible for the screening of more than one-third of all newborns in Taiwan [10]. In 2005, we were the first to implement Pompe disease newborn screening [1]. Initially, dried blood spot (DBS) GAA activities were measured using fluorogenic (4-methylumbelliferone) substrates [4]. Since 2015, we have been using tandem mass assay (MS/MS) substrates in order to accommodate multiplexing ability [11]. The medical genetics department in the NTUH is also the referral center for Pompe disease detected by the NTUH. The hospital staff work closely with the screening center in order to make a timely management of IOPD. Here, we describe our practice in the past seven years. 

## 2. Methods

NBS for inborn errors of metabolism in Taiwan was established in 1985, and, currently, the National Taiwan University Hospital (NTUH) holds one of the three screening centers in Taiwan. There are more than 300 birthing facilities that collect newborn dried bloodspots (DBS) and ship them promptly to the NBS labs. DBS sampling is usually performed 48–72 h after the birth of the babies, and shipping by priority mail typically takes less than two days. The NBS labs are requested to report high-risk results within 72 h after receiving the samples [10]. Although NBS is not mandatory in Taiwan, close to 100% of newborns acquired NBS. In 2008, Pompe disease newborn screening was added by the NTUH NBS Center and also by the other two screening centers [12], but written consent from the parent(s) is required. More than 95% of parents receiving service of NTUH NBS center provide consents for having Pompe disease newborn screening. The methods of Pompe disease NBS has been described previously [1,11,13]. Initially, GAA activity in DBS elute is measured using fluorogenic substrates, but the method was changed to the tandem mass spectrometry (MS/MS) at the end of 2015. From the first DBS, we set two cutoffs of the GAA activity measurements. Values exceeding the critical cutoff imply a high risk of having Pompe disease and that emergent confirmatory diagnostic testing is necessary. Values exceeding the borderline cutoff, mostly due to pseudodeficiency, will trigger a second-tier test, generating the value of % inhibition, before the final assignment [4]. If the second-tier test is positive, the baby will be suggested to have the confirmatory diagnostic testing. Since 2013, we employed the second-tier test to avoid requesting a second sample to prevent delay in the initiation of treatment. DBS DNA genotyping [14] may be applied to categorise the newborns. For confirmatory diagnostic testing, a whole blood sample was used for the measurement of GAA activity, genotyping, and CRIM test [10]. The CRIM test was performed by Western Blot analysis, using anti-GAA and anti-alpha-tubulin antibodies. For GAA protein detection, 15 μL of sonicated lymphocytes protein from patients was loaded in each lane (only 2 μL was used for normal control) and the X-ray film was exposed overnight. For alpha-tubulin (the control protein), 10 μL of sonicated protein was loaded in each lane and the X-ray film was exposed for 15 min. The Taiwanese common variant, p.D645E (p.Asp645Glu) [6,15], was rapidly screened by polymerase chain reaction-restriction fragment length polymorphism analysis (RFLP) using the *Bsa*HI restriction enzyme. Since p.D645E is a CRIM-positive variant [16], patients with this variant should be CRIM-positive. Enzyme replacement therapy (ERT) is scheduled for the next day after the heart involvement is confirmed, unless immunomodulation therapy to prevent anti-GAA antibodies production is planned. For babies without heart involvement at birth, a follow-up plan was initiated, including the development milestone, motor function, and biomarkers as described [10], and ERT was initiated until abnormalities appeared in the follow-up period.

## 3. Results

### 3.1. Performance of Screening and Diagnostic Testing

From 2016 to 2019, Pompe disease NBS was performed by the MS/MS method. The timeliness of the performance of other screening conditions (glucose-6-phosphate dehydrogenase deficiency, congenital hypothyroidism, galactosemia, congenital adrenal hyperplasia, and MS/MS acylcarnitine profile) during this period was similar to the previous three years (2013–2015); i.e., the compliance rates of reporting NBS results by 8 days of age were between 98.66% to 99.01%, compared to 98.85%–99.14% in the previous 3 years. With the MS/MS platform, 14 newborns were reported as high-risk for Pompe disease at a median age of 9 days (range 6–13). For newborns with a borderline risk for Pompe disease, 18 were reported at a median age of 13 days (9–28), but none of them were IOPD. During the previous 3 years (2013–2015), when Pompe disease NBS was performed using the 4MU platform, there were no Pompe disease high-risk newborns and 44 were reported as a borderline risk reported at a median age of 12 days (7–41), but none were IOPD.

Among the 14 newborns at a high risk of Pompe disease, four were found to have cardiomyopathy, as shown by electrocardiography, chest X-ray, echocardiography, and an elevation of serum creatine kinase (CK) and pro-brain natriuretic peptide (pro-BNP). Six [11,13] were found to have GAA deficiency and biallelic GAA variants but normal CK and no cardiomegaly and therefore were classified as potential LOPDs. The remaining four of the total 14 newborns were not affected. As for the 18 infants with the borderline risk, only 2 infants [11] were classified as potential LOPDs, while the rest were not affected. The four classic IOPD newborns were treated starting from a median of 9 days (8–14 days). *GAA* gene sequencing confirmed all pathogenic variants in these patients. Three of the four had at least one allele of the p.D645E variant. CRIM status (all CRIM positive (denoted as CRIM +)) was approved by Western Blot analysis in all four patients, including the one who did not have the p.D645E variation. The newborn who did not have the p.D645E variant did create a dilemma in the decision, and the history is described below. 

### 3.2. Case Description

A 7-days-old female newborn was requested to visit our hospital due to an abnormal Pompe disease screening result [13]. She was born at full-term to a G1P1 mother with a birth bodyweight of 4380 gm. The parents denied poor feeding, poor activity, nor weak crying in this baby. NBS included Pompe disease, and other conditions were performed on her third day of life. On Day 4, our NBS laboratory received her sample. On Day 6, a high risk of Pompe disease was reported (GAA activity 0.18 uM/h (critical cutoff < 0.5); ratio 42.87 (critical cutoff acid β-glucosidase (ABG)/GAA ≥ 20) so an urgent visit to our hospital was arranged on Day 7. When we saw her, she had normal muscle power, normal reflex, no macroglossia, but her facial folds decrease slightly. Laboratory examination revealed an elevation of pro-BNP (8738 pg/mL), CK (722 U/L), and alanine aminotransferase (ALT) (112 U/L). A chest X-ray revealed mild cardiomegaly (Figure 1). Echocardiography revealed moderate left ventricle (LV) and right ventricle (RV) hypertrophy, with a LV mass index (LVMI, measured by 2-D method) of 115.7 g/m^2^ (normal range < 65 g/m^2^). A whole blood sampling at Day 7 revealed deficient lymphocyte GAA activity (1.33 nmol/g pro/h, normal mean 66.7) and thus confirmed the diagnosis of IOPD. However, she did not have the common CRIM-positive Taiwan Pompe disease p.D645E variant, tested using DNA extracted from the first DBS. Although her CRIM status was unknown, her parents refused prophylactic immune modulation therapy. Western Blot analysis using the white blood cells as the material soon revealed the 110 kDa precursor GAA band (Figure 2), suggesting a CRIM-positive status. She received her first dose of rhGAA (20 mg/kg) at the age of 8 days. Mutation analysis showed heterozygous c.2024_2026del (p.N675del) and c.2040+1G>T variants *in trans*, compatible with IOPD. There were no CRIM status predictions about these two variants [9]. 

The study was approved by the ethical committee of National Taiwan University Hospital, Taipei, Taiwan (201906053RINB, 1st approved date 2019/08/05). The clinical information was gathered from the hospital medical records retrospectively, and no individual's consent was required.

## 4. Discussion

Pompe newborn screening was included in Taiwan’s newborn screening system as well as in the Recommended Uniform Screening Panel (RUSP) in the USA. Therefore, the timeliness requirements of newborn screening, i.e., the efficient collection, transportation, testing, and reporting of the results, also benefit Pompe newborn screening. In Taiwan, the recommended timelines for the high-risk babies is to report and communicate the results to the newborn’s healthcare provider/parent(s) within 6–8 days of life, regardless of the location of newborns, which maybe 300 km away from the screening centers/treatment centers (Figure 3). The compliance rate for reporting by age 8 days in our center was over 99%. In the present case, we informed the result by D6 and made the diagnosis by D7, demonstrating the discrimination power of applying the critical cutoffs and the well-established newborn screening system in Taiwan.

The more challenging part of this case was the preparation of ERT, starting after the confirmation of cardiomegaly and muscle damage. In this case, the decision for ERT was tentatively made at D7 and the ERT was initiated at D8, after reconfirming the GAA deficiency and confirming the CRIM status. In such classic IOPD newborns waiting for the decision of prophylaxis immunomodulation, we routinely check the predicted CRIM status by screening the p.D645E, commonly seen in our Pompe patients [6], using RFLP so that we could have the result in half day. Since p.D645E is related to CRIM-positive status [16], patients with at least one p.D645E allele will be CRIM-positive, and doing prophylaxis immunomodulation on such cases may not achieve benefit-risk balance. On the other hand, we plan to apply prophylactic immune modulation therapy for IOPD infants if a CRIM-negative status is confirmed. Therefore, in this case, we performed the blood Western Blotting assay as described [17] to determine the CRIM status since the GAA sequencing result took more time, and the CRIM status for a novel variant may not be predicable, especially for the splicing mutation [9] presented in this case. Rapid sequencing, or screening for several common variants, may replace the blood CRIM status measurement in such a situation. 

In conclusion, we demonstrate here the performance of the NBS system in Taiwan and the decision steps for positive Pompe NBS. Knowing the genotype/CRIM status was necessary for the ERT initiation but it made for very intensive work. Depending on the different geographic regions and various resources available, each team needs to prepare themselves with a standardized and comprehensive algorithm for the confirmation and treatment of classic IOPD patients. With the timeliness of screening and diagnosis, we were able to start the treatment as early as possible to achieve the best treatment outcome. 

## Figures and Tables

**Figure 1 IJNS-06-00030-f001:**
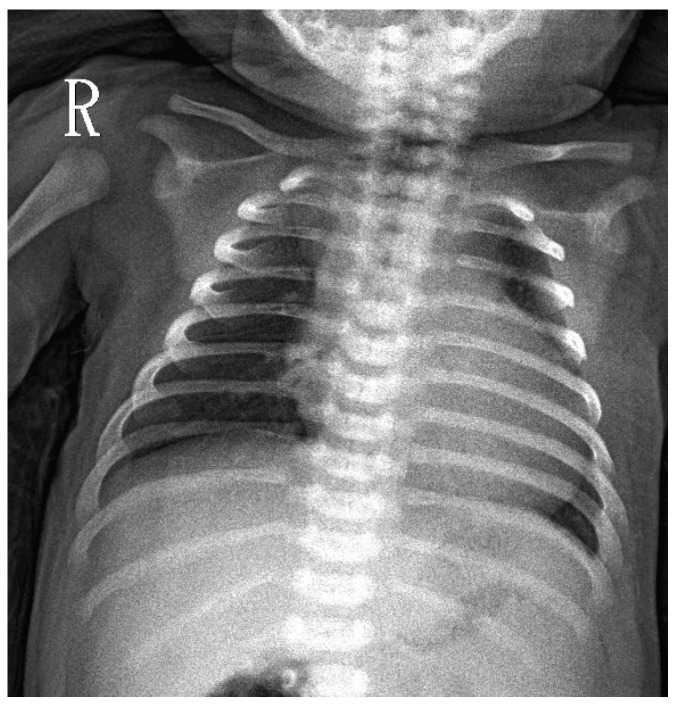
CXR at D7 in a newborn with a positive Pompe newborn screening result. Mild cardiomegaly was noted.

**Figure 2 IJNS-06-00030-f002:**
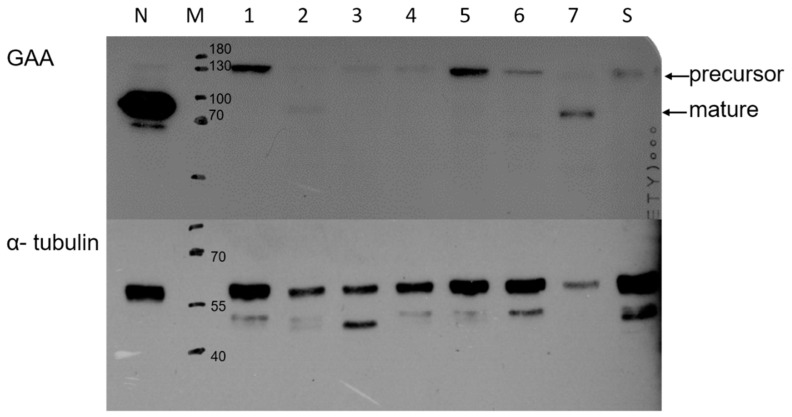
Blood glucosidase alpha acid (GAA) Western Blotting of this case (S), indicating a cross-reactive immunologic material (CRIM)-positive status. The lymphocytes were sonicated and then blotted with antibodies to detect human GAA and α-tubulin protein presence. M: marker; N: normal newborn; No. 1 and No. 3 were from infantile-onset Pompe disease (IOPD) patients with the CRIM(+) GAA variant; No. 2, No. 4–7 were from newborns with low GAA activity. N: normal control sample; M: marker. Precursor: 110 kDa GAA. Mature: 70/76 kDa GAA.

**Figure 3 IJNS-06-00030-f003:**
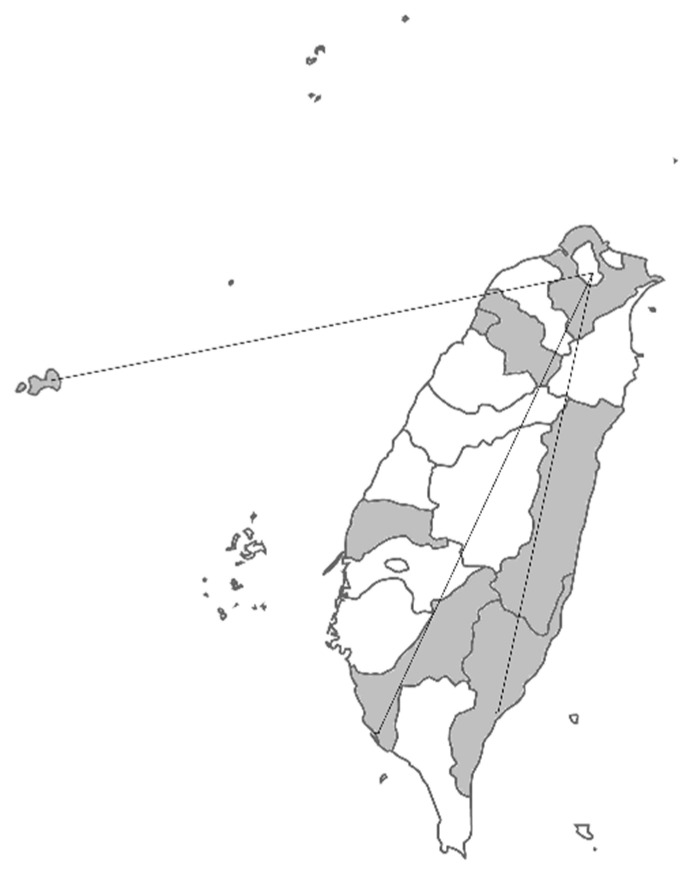
The coverage map of the National Taiwan University Hospital Newborn Screening Center. The samples ship from the gray areas to our screening center, designated by the government, including the three far areas indicated by the lines. The distances from the birthplaces to the screening center were around 300 km.

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
