# Peer review of "The Timely Needs for Infantile Onset Pompe Disease Newborn Screening—Practice in Taiwan"

_2409-515X, 2020, doi:10.3390/ijns6020030_

Round 1
Reviewer 1 Report
It would enhance the manuscript for readers who are not intimately familiar with Taiwan to provide some information about how many other centers receive Pompe samples, and how they process them, and what metrics do they use. Are they similar or different than the ones used at the National Taiwan University Hospital?
It is unclear to this reviewer whether any additional sequences were identified which would predict whether the mutations seen are CRIM + or CRIM -. Could you comment on this?
It seems that the determination of CRIM status takes considerable time from the text. Why is that?
Are virtually all babies currently born in Taiwan being screened for Pompe disease?
How are you following the patients who are not iimmediately recognized as infantile. Some of the later onset patients identified elsewhere have quite early onset disease. You should comment about this.
Author Response
Reviewer comments
Reviewer 1
Comments and Suggestions for Authors
It would enhance the manuscript for readers who are not intimately familiar with Taiwan to provide some information about how many other centers receive Pompe samples, and how they process them, and what metrics do they use. Are they similar or different than the ones used at the National Taiwan University Hospital?
Answer:
Thank you for your instruction. In total, there are three NBS centers in Taiwan, and all provide pompe NBS using different methods and algorithm. We here only share our practice to demonstrate the complexity and the solution under careful consideration.
Line 52: NTUH hosts one of the three screening centers in Taiwan.
Line 58: A reference for the other two NBS centers were added.
It is unclear to this reviewer whether any additional sequences were identified which would predict whether the mutations seen are CRIM + or CRIM -. Could you comment on this?
Answer: Thank you for your remind. This has been explained further in the method as “Since D645E is a CRIM-positive variant [16], patients with this variant should be CRIM-positive.” (Line 77-79). In addition, a comment has been made in the discussion as “and the CRIM status for a novel variant may not be predicable especially for the splicing mutation [9] as presented in this case. “ (Line 158-159).
It seems that the determination of CRIM status takes considerable time from the text. Why is that?
Answer: No, the CRIM status now is based on the blood assay, not the fibroblast assay as in the old literatures. Therefore the time for CRIM status determination could be as short as half day, as described in this paper.
Are virtually all babies currently born in Taiwan being screened for Pompe disease?
Answer: Although a consent is required, most parents provide consent for Pompe NBS in Taiwan. Added the description in method (Line 59).
How are you following the patients who are not iimmediately recognized as infantile. Some of the later onset patients identified elsewhere have quite early onset disease. You should comment about this.
Answer: Yes, we follow those infants with potentially LOPD. However, this report aims discussing the timeliness for IOPD. Therefore we only add the description for those screening positive newborns (as NBS-LOPD) in the results (Line 81-83).
Reviewer 2 Report
Thank you for the opportunity to review this paper about the recent history of screening for Pompe disease in Taiwan using MS/MS. With the expansion of screening for Pompe disease, it is important to hear the experience of programs with significant history in screening.
Specific comments:
- In the methods section - the second tier test is not described. Please include details about the second tier test, or a quick summary and a reference.
- Some non-standard abbreviations are used, but not defined (ABG...)
- there is a typo on line 85 (Pome --> Pompe)
- Consider switching mutation --> variant to align with current recommendations
- A flow chart of the confirmatory testing protocol (or a reference to a previously published version) would be helpful
- What was the eventual outcome of the 9 "high risk" infants who did not have IOPD and the 18 "borderline"?
- There is an extra period in the caption for Figure 2.
- A table summarizing the results of all screen positive patients (or true positive patients) would be helpful for the results section (age, bw, nbs result, age at report, at at treatment, confirmatory testing, variants, etc)
- Was there any published information / prediction available about the CRIM status for the patient described in detail here?
Author Response
Reviewer comments
Reviewer 2
Thank you for the opportunity to review this paper about the recent history of screening for Pompe disease in Taiwan using MS/MS. With the expansion of screening for Pompe disease, it is important to hear the experience of programs with significant history in screening.
Specific comments:
In the methods section - the second tier test is not described. Please include details about the second tier test, or a quick summary and a reference.
Answer: added (Line 66)
Some non-standard abbreviations are used, but not defined (ABG...)
Answer: Thank you for your remind and we have checked the manuscript and revised all the abbreviations.
there is a typo on line 85 (Pome --> Pompe)
Answer: revised
Consider switching mutation --> variant to align with current recommendations
Answer: Thank you for your remind and we have checked the manuscript and revised all the terms.
A flow chart of the confirmatory testing protocol (or a reference to a previously published version) would be helpful
Answer: added the reference (Line 71, ref 13)
What was the eventual outcome of the 9 "high risk" infants who did not have IOPD and the 18 "borderline"?
Answer: The distribution for the high risk infants (n=14) was IOPD (n=4), LOPD (n=6), and negative (n=4). The distribution for the borderline risk infants (n=18) was IOPD (n=0), LOPD (n=2), and negative (n=16).
The information has been added in the result as “Six [11,13] were found to have GAA deficiency and biallelic GAA variants, but normal CK and no cardiomegaly and therefore were classified as potential LOPD. The rest 4 of the total 14 newborns were not affected. As for the 18 infants with the borderline risk, only 2 infants [11] were classified as potential LOPD while the rest were not affected. “(Line 99-102)
There is an extra period in the caption for Figure 2.
A table summarizing the results of all screen positive patients (or true positive patients) would be helpful for the results section (age, bw, nbs result, age at report, at treatment, confirmatory testing, variants, etc)
Answer:
Thank you for your suggestion. However, we decided not adding this summary table for 2 reasons:
- The aim of this report is targeting the timeliness, including the implantation of the critical cutoffs and the associated CRIM status and genotype analysis.
- The information of the true positives have been reported previously [11, 13].
Therefore we add has been added in the result as “Six [11,13] were found to have GAA deficiency and biallelic GAA variants, but normal CK and no cardiomegaly and therefore were classified as potential LOPD. The rest 4 of the total 14 newborns were not affected. As for the 18 infants with the borderline risk, only 2 infants [11] were classified as potential LOPD while the rest were not affected. “(Line 99-102)
Was there any published information / prediction available about the CRIM status for the patient described in detail here?
Answer: No, there are no information regarding to the two variants. It therefore is important to have the ability to determine CRIM status in such situation. (Added in results, Line 126)